# 3D-Printed Microneedles for Point-of-Care Biosensing Applications

**DOI:** 10.3390/mi13071099

**Published:** 2022-07-13

**Authors:** Misagh Rezapour Sarabi, Sattar Akbari Nakhjavani, Savas Tasoglu

**Affiliations:** 1Mechanical Engineering Department, School of Engineering, Koç University, Istanbul 34450, Turkey; msarabi19@ku.edu.tr (M.R.S.); sakbarinakhjavani@ku.edu.tr (S.A.N.); 2Koç University Translational Medicine Research Center (KUTTAM), Koç University, Istanbul 34450, Turkey; 3Physical Intelligence Department, Max Planck Institute for Intelligent Systems, 70569 Stuttgart, Germany; 4Koç University Arçelik Research Center for Creative Industries (KUAR), Koç University, Istanbul 34450, Turkey; 5Boğaziçi Institute of Biomedical Engineering, Boğaziçi University, Istanbul 34684, Turkey; 6Koç University İş Bank Artificial Intelligence Lab (KUIS AI Lab), Koç University, Sariyer, Istanbul 34450, Turkey

**Keywords:** microneedles, biosensing, 3D printing, point-of-care

## Abstract

Microneedles (MNs) are an emerging technology for user-friendly and minimally invasive injection, offering less pain and lower tissue damage in comparison to conventional needles. With their ability to extract body fluids, MNs are among the convenient candidates for developing biosensing setups, where target molecules/biomarkers are detected by the biosensor using the sample collected with the MNs. Herein, we discuss the 3D printing of microneedle arrays (MNAs) toward enabling point-of-care (POC) biosensing applications.

## 1. 3D Printing for Fabrication of Microneedle Arrays (MNAs)

The organization of needles in micro- and nano- scales on an array or patch, introduced the emerging term of microneedle arrays (MNAs), are prominent minimally invasive injection setups for medical and biomedical applications ranging from delivery of drugs and agents to body to extracting body fluids and bio signal acquisition, to name a few [1]. Minimizing insertion pain in comparison with conventional needle technologies, MNAs not only offer a solution for needle phobia, the anxiety linked to the injection or utilizing needles, but also pave the way for the fabrication of more convenient devices that decrease tissue damage [2]. On the other hand, while conventional needles are usually made of stainless steel, MNAs can be fabricated with a variety of alternatives such as polymers and inorganic materials in various geometries, with different heights, diameters, and needle shapes [3]. Also, based on the end application of MNAs, they can be prepared in solid, coated, hollow, or biodegradable/dissolvable forms.

Over the past decade, the fabrication method of MNAs has been advancing from molding, micro machining, and photolithography to the emerging method of 3D printing, where the structure is produced in a layer-by-layer manner using a computer-aided design (CAD) input [4]. 3D printing can be carried out using various technologies as its working mechanism, including, but not limited, to stereolithography (SLA), digital light processing (DLP), two- or multi- photon polymerization (TPP or MPP), fused deposition modeling (FDM), selective laser sintering (SLS), and selective laser melting (SLM). While conventional fabrication methods require cost- and time- consuming procedures of high labor, cleanroom facilities, and/or unusable material waste, 3D printing is an additive manufacturing method that can be used directly for fabrication of MNAs, offering ease of access, being capable of rapid prototyping, and enabling cell-encapsulated structures with integration with bioinks. For example, fabrication with lithography protocol in a cleanroom not only needs expensive facilities that are usually not highly accessible but also requires several steps such as wafer washing and preparation, applying a photoresist coat, soft baking, exposing, post-exposure baking, and development. On the other hand, lithography is not a convenient approach for rapid prototyping; i.e., all the steps should be retaken if only one feature of the product is changed. 3D printing, with comparatively high accessibility and a higher level of reproducibility, offers a solution to these challenges [5]. Furthermore, since there is no requisite for expensive consumables such as wafers and photoresists, 3D printing reinforces its place for a cost- and time-efficient fabrication approach for MNA-based biosensors. 3D printing methods have been used for fabrication of immense pieces such as automotive parts and turbines, as well as minute delicate objects such as vessel stents and MNAs. For the case of such delicate items, high-precision 3D printing is required to be able to print the tiny features in the desired form. Precision is usually enabled in 3D printers by smaller layer heights, optimized printing orientation, smaller spot diameter, optimized laser power, and more accurate nozzles. 3D-printed MNAs hence are emerging for fabrication of advanced healthcare setups [6], enabling personalized treatment options and point-of-care (POC) diagnosis, i.e., testing at or near the clinical point of patient care Figure 1.

## 2. MNA-Based Point-of-Care (POC) Biosensing

Advancements in nanotechnology and biosensing sciences have facilitated utilizing POC biosensing platforms in various fields, including medicine, industry, and defense [8,9]. A biosensor is composed of two fundamental parts: a bio-recognition element and a signal transducer [10]. Applying specific biomolecules in the presence of noble nanostructures (NSs) and proper signal transducers has caused POC biosensing platforms to demonstrate enhanced performance in terms of limit of detection (LOD). Due to the outstanding properties that NSs offer, such as increasing active surface area and enhancing conductivity, they have successfully been utilized for development of POC biosensing platforms [11]. In addition, different biosensing platforms have been designed for the accurate detection of various diseases, cancer biomarkers, drug residues, environmental pollutants, and pesticides [12]. Among electrochemical, fluorescent, luminescent, piezoelectric, and quartz crystal microbalance transducers, which have been applied for development of POC biosensing platforms, the electrochemical approach has attracted higher attention due to its sensitivity and simplicity. Taking advantage of the largest organ of the human body, skin, the integration of MNAs with biosensor technologies will provide new insights into POC biosensing applications. Additionally, modifying MNAs with specific bio-recognition elements such as aptamer can dramatically elevate the performance of the MNAs-based sensing platform. For example, hydrogel MNs were modified with a fluorescent-labeled aptamer probe for detection of glucose in a model animal, with minimum invasiveness, great specificity, and sensitivity [13]. Furthermore, another study presented a MN-supported electrochemical, aptamer-based sensor for continuous drug monitoring [14]. Conventional methods of MN sensing could only sense molecules experiencing redox reactions, limiting their capability to detect non-redox-active metabolites, biomarkers, and therapeutics. The proposed aptamer-based MN-based sensor overcame this barrier and offered real-time electrochemical detection. To reveal this capability, the fabricated biosensing platform was examined every 4.5 s, while flowing buffer at a rate of 0.66 mL/min.

Generally, MNA-based biosensing platforms utilize two main approaches for biomarker detection, i.e., direct and indirect detection. In brief, during the direct detection approach, whole procedures, including the sample obtaining and recognition of corresponding biomarkers and readout, occur in an all-in-one platform, while in the indirect approach, MNs are used for sampling and the detection procedure occurs in another instrument. It is worth mentioning that fabrication of the direct MNA-based biosensors needs various technologies such as microfluidics, nanotechnology and microfabrication which could be considered as a disadvantage regarding their application. However, their advantages exceed such limitations.

One of the main purposes for developing MNA-based biosensors is providing a pain-free, safe, and convenient sampling procedure, followed by the detection of desired biomarkers with the minimum amount of invasion. In this regard, one MNA-based biosensor was designed using an ultra-sensitive fluorescent optic fiber for transdermal glucose monitoring in which the glucose binding protein was immobilized on Nickel-Nitriloacetic acid agarose beads [15]. The obtained results revealed that the developed biosensing system would be a promising device for noninvasive glucose monitoring. Furthermore, continuous glucose monitoring (CGM) plays a promising role in the management of diabetic patients by providing real-time data of body glucose concentrations. Considering the exceptional correlation of blood glucose and ISF glucose concentrations, a MNA was microfabricated using DLP 3D printing technology for CGM applications [16]. Followed by microfabrication, electrochemical sensing electrodes including work, reference, and counter were produced by the electroplating of MNAs (Figure 2A). The biorecognition element, the glucose oxidase (GOD) enzyme, was immobilized on the desired surface of the working electrodes. Upon successful assembly of the 3D-printed biosensing platform, the setup was employed to measure the subcutaneous glucose of the dermis layer of a mouse skin (Figure 2B). The comparison of the obtained results of this platform for *in vivo* monitoring of subcutaneous glucose in a normal mouse and in a diabetic mouse with a commercial glucometer demonstrated the accuracy of the device, facilitating future *in vivo* CGM studies (Figure 2C,D).

Also, a MN-based biosensor was designed for the continuous monitoring of ketone bodies, utilizing an electrochemical technique for detection of β-hydroxybutyrate (HB), as an indicator of ketones. The developed biosensor demonstrated sensitive selectivity for the detection of target molecules in the artificial interstitial fluid (ISF). It also presented the ability for the detection of HB besides glucose and lactate, which could be considered as a simultaneous MNA-based biosensor with the capability of sensing multiple biomarkers of diabetic ketosis [17]. Additionally, transdermal drug delivery has attracted the attention of physicians due to its promising advantages compared to oral administration, such as minimum side effects, gastrointestinal bleeding, and problems regarding liver metabolism. As a promising application, a 3D-printed microheater sensor-integrated MNA was designed for pain management in the patients with lower back pains [7]. A drug-encapsulated MN platform was integrated with a 3D-printed microheater for precise control of drug delivery in the injured area. A printing ink solution of multiwalled carbon nanotubes (MWCNTs) and polydimethylsiloxane (PDMS) was employed to microfabricate the microheater Figure 2E, and then it was capitalized on the back surface of the MNAs. The setup was used for the in vitro controlled release of the model drug of near-infrared fluorescence dye Cy5 in mouse skin (Figure 2F,G). The results of in vitro investigation drug delivery application of the developed platform facilitate a new way for application of MNA-integrated biosensing platforms in transdermal drug delivery and sensor-controlled medical systems.

During recent years, the development of MNA-based biosensors for the real-time detection/monitoring of various diseases such as cancer and diabetes has widely been investigated. Considering biofluids such as blood and ISF as rich sources of biomarkers, such MNA-based biosensors may be employed for both sample gathering and the read-out process [18]. To obtain accurate and convincing results in terms of determining various biomarkers using MNA-based biosensors, the application of nanomaterials such as gold and carbon nanotubes in their design and development is inevitable. The nano-coating of MNA not only improves the detection performance of biosensing platforms but also facilitates additional functionalization of MNA with bio-transducers such as antibodies and aptamers. It should be noted that the design of MNA patch in terms of size, material, geometry, and mechanical properties should also be considered in developing the MNA-based biosensors. Experimentally reported MNs with applications in sampling or extraction of body fluids such as ISF, blood, and glucose detection delineated the heights of MN to be in the range of 400 to 800 µm [2]. Also, commercial MN vaccine delivery platforms mostly have a height of 600 µm [19]. In addition, polymers with different degradation rates, swelling characteristics, and behaviors in response to biological and physical stimuli can be utilized to manufacture polymeric MNs with different mechanical properties and performance [20]. Conducting polymers can also enable the fabrication of more functional MNA-based biosensors by offering conductivity and modification capability.

A highly porous gold MNA-based biosensor was fabricated using electrochemical self-templating approach [21]. The MNA was microfabricated on a polycarbonate scaffold. After electrochemically cleaning the MNA, the high-porous gold was electrodeposited on the surface of working electrodes. After stabilizing, washing, and activating the electrodes, the as-prepared highly porous MNA was modified with the acting enzyme glucose dehydrogenase (GDH). The characterization results revealed that fabricated MNA had a larger active surface area compared to bare MNA (100 times larger). As a concept of application, the designed system was applied for glucose monitoring in the artificial ISF in a skin model consisting of chitosan/agarose hydrogels. The performance of the designed MNA-based biosensor in terms of sensitivity, stability, and selectivity was evaluated satisfactory.

Skin ISF is a biofluid containing information-rich biomarkers which can be utilized for several purposes such as diagnosis and prognosis of diseases, analysis of tumors, and drug inspection. Hence, 3D printing MNAs for ISF-collection and -analysis biosensors can enable a great opportunity for developing POC systems. An attempt in this regard took advantage of photopolymerization 3D printing technology for the fabrication of MNAs with open groove channels, which were aimed to be used for extraction of body liquids [22]. The groove structure was designed for enabling fluid extraction with the help of capillary forces. The 3D-printed MNAs were tested on porcine skin and several parts of chicken tissues for their skin piercing capability. Afterwards, for demonstrating MNAs’ ability for fluid extraction and biomarker analysis, an in vitro hydrogel model was used. This model mimicked skin and contained 1% agarose hydrogel premixed with several values of glucose ranging from 0.1 to 2 wt%. Commercial glucose strips were attached to the backside of the MNAs, and the colorimetric response of the strips were observed for the amount of the glucose that MNAs were capable of extraction within a period of 60 s. The results demonstrated convenient detection sensitivity and exhibited a setup for rapid detection upon extraction.

Thanks to the MNAs, various studies have initiated to evaluate and investigate the ISF-originated biomarkers such as nucleic acids, proteins, and small molecules. Compared to the other biofluids such as blood, saliva, or urine, these biomarkers could be easily obtained. Organophosphorus nerve agents (OPNAs) are one of deadliest chemical ever produced for warfare application. Skin is one of the sites that could extensively be exposed to such chemicals. In this regard, an electrochemical MNA-based biosensor was designed for rapid, sensitive, and specific detection of OPNAs from ISF [23]. Having used the transdermal drug delivery applications of MNAs, a hollow MNA with a pyramidal shape was fabricated using acrylate-based polymer as main material. Following the preparation of the MNAs, they were treated and covered with carbon paste to enhance their performance toward detecting OPNAs. It worth mentioning that 2 out of every 3 MNs were modified with carbon paste, while the third one in each array was modified with Ag/AgCl ink (to serve as the refence electrode). After the successful assembly of MNAs and their connectors, the hollow MNA-based biosensor was applied for detection of target analyte in a detection range of 20 to 180 µM. Furthermore, the applicability of the fabricated MNA-based biosensor was investigated by its application into a mouse skin that was previously exposed to OPNAs. The developed platform presented a great performance for the *in vivo* detection of the target analyte (with an effective recovery). The developed platform was suggested as a potential approach for the further development of on-body measurement.

For the purpose of sequence-specific sampling, isolation, and the detection of nucleic acid biomarkers from skin ISF, MNs coated with an alginate–peptide nucleic acid hybrid material were developed [24]. The device was demonstrated for 15 min, pressing on the skin for sufficient sampling. MNAs were functionalized with bespoke peptide nucleic acid (PNA) probes, which were covalently bound to an alginate hydrogel matrix via a photocleavable linker (PCL). The alginate-azide polymer was prepared as by ethyl (dimethylamino propyl) carbodiimide/N-hydroxysuccinimide (EDC/NHS) mediated peptide coupling between low-viscosity alginate and 11-azido-3,6,9-trioxyundecan-1-amine. All three hydrogel samples (unmodified alginate, alginate azide, and alginate-PNA) were characterized. The nuclear magnetic resonance (NMR) spectrum of alginate-azide was analyzed to characterize the amount of azide functionalization. The results demonstrated that isolating miRNA biomarkers efficiency is higher for alginate-PNA, alginate azide, and unmodified alginate, respectively.

Surface-enhanced Raman spectroscopy (SERS) is a robust method that has been applied for sensitive detection of various molecules based on Raman scattering of target analyte on a plasmonic surface. Considering the unique molecular specificity of Raman spectroscopy, so called biomolecule fingerprinting, SERS based biosensors could be employed for development of label-free POC MNA-based biosensors. In this regard, a biocompatible MNA-based sensor was fabricated for accurate detection of methylene blue (MB) as model analyte in mouse skin [25]. The histologic results of mouse skin presented insignificant immune response regarding insertion demonstrating the minimally invasiveness of MNA. The performance of the fabricated system for the detection of MB was calculated for 1 nM via a single insertion/withdraw, suggesting its potential application for developing label-free intradermal MNA-based biosensors. Advances in producing SERS substrates have attracted biomedical engineers to develop enhanced MNA-based biosensors for detection of various analytes such as pH, glucose, and alcohol using ISF as a pool of biomarkers. The Norland optical adhesive 65 polymer (NOA 65) was used for the fabrication of MNA using a commercial PDMS mold [26]. Due to the presence of mercaptoester groups and the transparency of the material on the other hand, gold nanorods could be functionalize on the surface of MNAs and hence, the collecting of SERS signals could be facilitated. The results showed that the fabricated MNA-based sensor could effectively quantify pH over a range of 5 to 9. Interestingly, and as proof of application, the fabricated system successfully detected pH levels in the artificial agar gel skin phantom and human skin in situ with a good robustness and proper stability even after one-month incubation in phosphate-buffered saline.

## 3. Outlook and Challenges

Biocompatibility is one of the main prerequisites for the devices that are in close contact with human body, which means that such devices should not cause any adverse effect or attract responsive behavior from the immune system. This highlights that MNA-based biosensors that are proposed for translation and commercialization purposes considering their benefits should satisfy biocompatibility requirements as well. While using biocompatible materials in the fabrication of MNA-based biosensors is an obvious solution to achieve this goal, immunomodulatory coatings can be used as an alternative approach when there is a limitation for choosing biocompatible materials. Aimed to decrease triggering of the immune system of the human body and promote integration with the surrounding organs, immunomodulatory coatings can be applied on the needles in the same way that a drug is coated. On the other hand, in the case that biodegradable materials are used in the fabrication procedure, residues of the degradation should be harmless to human body as well. In this regard, polycaprolactone (PCL) and polylactic acid (PLA) are among the examples that have been studied to be not only biocompatible but also not leaving behind any irritating degradation products [6]. Another main concern regarding the translation to clinical practices is that most of the available reports of MNA-based biosensing procedures have been performed within phantoms that are mimicking human tissues or on animals. Utilizing organoids can be a helpful approach here to increase the pace of translation since organoids offer more realistic comprehension of the human body in comparison with the conventional methods.

Analysis of chemical data in biosensor systems usually requires several chemometrics methods dealing with mathematical and statistical models followed by optimization procedures, which can be time-consuming in systems that are aimed for smart rapid detection and/or sensing various parameters at once. One of the approaches to enable more advanced POC robust biosensor setups in the future can be enabled by the incorporation of various detection/sensing frameworks along with several analysis layouts. In this regard, the integration of artificial intelligence (AI) techniques, for instance, machine learning (ML) and deep learning (DL), can be a useful approach for big data analysis and automation of sensing. On the other hand, AI techniques can contribute to fabricating steps as well; for example, features of 3D-printed MNAs were reported to be optimized and predicted using ML and DL [27]. On the other hand, the information from different biomarkers, aptamers, and sensing materials can be utilized to create libraries for ML training models, which can pave the way for predictive systems, resulting in development of more advanced healthcare setups [28]. Furthermore, 3D printing is an emerging scalable fabrication method and hence can prepare the infrastructure for manufacturing and commercializing setups integrating MNAs with biosensors.

Advances in the development of non-invasive MNA-based biosensing platforms for monitoring of various biomarkers is an interesting area in the field of biosensors. Considering the great potential of electrochemical and SERS techniques, the development and commercialization of MNA-based POC biosensors for the multianalyte detection of biomarkers would be an asset in the management and monitoring of different diseases. Interestingly, such biosensing devices would be proper candidates for developing wearable patches for the online monitoring of the desired biomarker with the minimum amount of invasiveness in the patients. For instance, the design and fabrication of MNA-based POC biosensing platforms (wearable patches) would be suggested for real-time monitoring of glucose in diabetic patients or sensing of lactic acid in athletes. Although many advances regarding the development and application of MNA-based biosensors for the detection of various biomarkers have been achieved, still, some challenges in terms of real-time monitoring of biomarkers remain unsolved. In addition, in MNAs with nonconductive materials, effectively modifying the 3D-printed MNA-based biosensors with nanomaterials would be a concern. In this regard, MNs fabricated with metals can be considered as the initial solution for conductivity, however, fabrication of metals usually requires more time- and cost-consuming procedures. On the other hand, 3D printers are usually based on inks made with material family of polymers. Progress in the science of polymers is advancing to the development of novel conducting polymers such as polyaniline (PANI), polypyrrole (PPy), polythiophene (PTh), and polyacetylene [29]. Hence, conducting polymers can introduce a new approach for fabrication of more functional MNA-based biosensors.

The other limitation that may affect the translation of MNA-based biosensing platforms from benches to beds would be their high cost of assembly processes, which need advanced laboratories. It is believed that further investigations should be performed to address the shortcomings associated with the application of 3D-printed MNA-based biosensing platforms. However, by the advancement of technology, particularly the development of innovative printing approaches and the application of affordable materials for microfabrication of MNAs, these biosensing platforms will smoothly find their place in the clinical trials and will be commercially available in the near future. Focusing on the number of clinical investigations of MNA-based biosensing platforms reveals that there is a demanding interest toward their development as new diagnostic techniques for detection of various biomarkers.

## 4. Conclusions

The advances in technologies will revolutionize the field of POC biosensing platforms in terms of convenience, safety, non-invasiveness, biocompatibility, cost effectiveness, and ease of fabrication. In other words, by entering into the smart devices era, a new group of MNA-based biosensing platforms will emerge. Such health biosensors are expected to provide a real-time analysis reflecting the condition of the applied environment. It is believed that by integrating smartphone technologies into the MNA-based POC biosensors, a new generation of these devices will act as a game-changer in the field of detection. Initially, the monitoring and evaluation of glucose and its derivatives was the main purpose of developing MNA-based POC biosensors. Next, by understanding their advantages, their application in detection of other types of biomarkers has been initiated. An evaluation on the growing number of published papers about MNA-based POC biosensors demonstrates the increasing interest toward designing these devices, which are at the early stages of their road and will find their main role in the near future.

## Figures and Tables

**Figure 1 micromachines-13-01099-f001:**
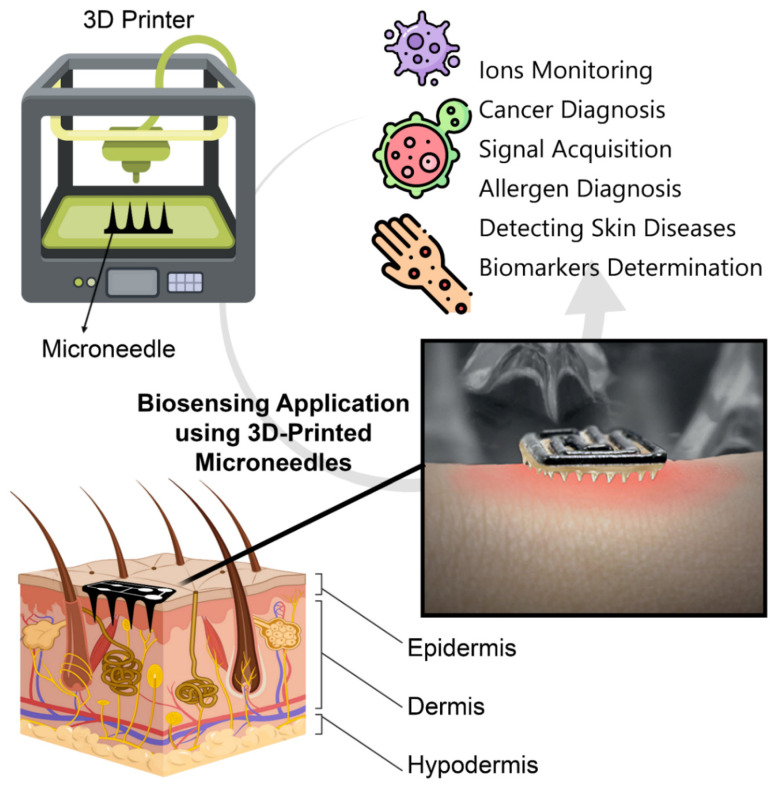
Integration of microneedle arrays (MNAs) with point-of-care (POC) biosensor technologies will provide new insights in biosensing applications. Taking advantage of emerging manufacturing method of 3D printing, MNA-integrated biosensors can be fabricated with customizable properties rapidly, resulting in biomedical applications such as monitoring different particles (for instance, biomarkers, and ions) and diagnosis of diseases (for instance, cancers, and allergies). The figure in the box was adapted from the journal cover photo of ref. [7] with permission from John Wiley & Sons (2019).

**Figure 2 micromachines-13-01099-f002:**
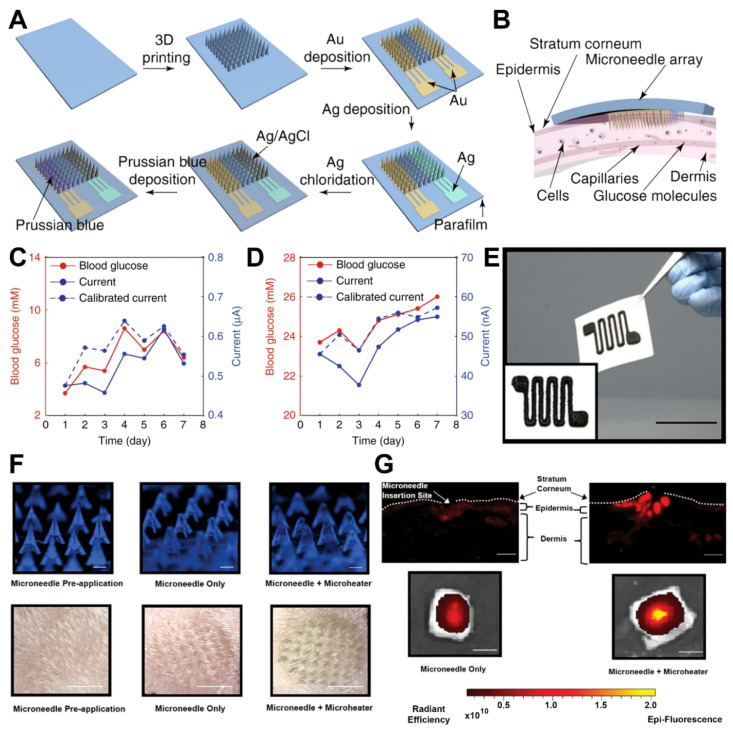
Examples of MNA-based biosensors. (**A**) Fabrication steps of the MN biosensing device for glucose monitoring. (**B**) Schematic representation of the device insertion into the skin. (**C**) The device was used for *in vivo* monitoring of subcutaneous glucose in a normal mouse and (**D**) in a diabetic mouse. In both diagrams, the mouse was monitored for seven days. The blue lines indicate the MN biosensing device, and the red lines indicate a commercial blood glucose meter. (**E**) A 3D-printed microheater sensor-integrated MN patch was developed for pain management application. Optical image of the microheater printed on a paper (Scale bar: 35 mm). (**F**) The setup was used for in vitro controlled release of the model drug of near-infrared fluorescence dye Cy5 in rat skin (Scale bars, top row: 500 µm, bottom row: 5 mm). (**G**) Fluorescence images representative of the rat skin tissue slides, indicating wider and deeper diffusion of Cy5 (upper row) and more initial drug depot in skin (lower row), in microheater integrated group in comparison with the MN-only group (Scale bars, top row: 100 µm, bottom row: 5 mm). Subfigures (**A**–**D**) were adapted with permission of Springer Nature (2021) from ref. [16], and subfigures (**E**–**G**) were adapted with permission of John Wiley & Sons (2019) from ref. [7].

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
