# Peer review of "3D-Printed Microneedles for Point-of-Care Biosensing Applications"

_micromachines, 2022, doi:10.3390/mi13071099_

Round 1

Reviewer 1 Report

This manuscript discusses 3D printing of microneedle arrays (MNAs) toward enabling point-of-care (POC) biosensing applications. 3D printing technology is a convenient technique to directly fabricate MNAs with high precision. Is this characteristic of high precision important for fabricating MNAs? Modifying MNAs as a biosensor can provide a painfree, safe, and convenient sampling procedure. Does this application need a specific fabrication process or MNAs material? In addition, the 3D printing technique in fabricating biosensor needs to be highlighted.

Author Response

We thank the reviewers for valuable suggestions to improve our manuscript. We addressed all the comments of the reviewers. Our point-by-point responses are listed inside the attached file.

Reviewer 2 Report

1. On page 1:“while conventional fabrication methods require cost- and time- consuming procedures of high labor, cleanroom facilities, and/or unusable material waste, 3D printing is an additive manufacturing method that can be used directly for fabrication of MNAs, offering ease of access, capable of rapid prototyping.” The authors may want to compare the complexity, time, and cost of MNAs with conventional fabrication methods.

2. The authors said that modifying MNAs with specific bio-recognition elements such as aptamer can dramatically elevate the performance of the MNAs-based sensing platform. It is suggested to list some comparisons between MNAs-based sensing platforms and conventional methods in terms of sampling volume, detection time, detection sensitivity, etc.

3. The authors mentioned MNA detections are almost all about glucose, I suggest complementing the application of the detection of other biomarkers (nucleic acids, proteins), which are important for the wide application of MNAs-based sensing platforms.

4. The words in the bottom image of Figure 2G are too small. I suggest using a clearer image in Figure 2G for easier reading and understanding.

5. In page 3: “ It should be noted that the design of MNA patch in terms of size, material, geometry, and mechanical properties should also be considered in developing the MNA-based biosensors.” However, the authors only cite an example of a highly porous gold MNA-based biosensor, and there are no descriptions of size, geometry, and mechanical properties, which are suggested to be added.

6. The authors said that chemical data analysis in biosensor systems usually requires several chemometric methods which can be time-consuming. Integration of artificial intelligence (Al) techniques for big data analysis and automated sensing can be a useful approach: it is not clear exactly what percent of time is saved by Al techniques, and are there already relevant applications to prove this point? Please cite the relevant literature.

Author Response

(The authors gave the same response as above.)
